# Glycemic control and bacterial infectious risk in type 2 diabetes: A retrospective cohort from a primary care database

Edouard Lemoine[1]*, Mikaël Dusenne[2]°, Matthieu Schuers[1,2,3]°

**1** Department of General Practice, UNIROUEN, Normandie Université, Rouen, France, **2** Department of Medical Information and Informatics, CHU Rouen, Rouen, France, **3** Medical Informatics and e-Health Knowledge Engineering Laboratory, INSERM, U1142, LIMICS, Sorbonne University, Paris, France

° These authors contributed equally to this work.
* edouard.lemoine1@univ-rouen.fr

**Data Availability Statement:** Access to aggregated raw data is possible through a request to PRIMEGE Normandie's ethical and scientific committee as described in the database's internal regulations, either by contacting the data manager at primege.

## Abstract

### Objective

The prevalence of diabetes was estimated at 5.3% of the French population in 2020. People with type 2 diabetes have an increased risk of infection. Currently, there is no consensus on the impact of glycemic control on infectious risk. The objective was to evaluate whether glycemic control and diabetes severity were associated with infectious risk in type 2 diabetes.

### Materials and methods

We designed a multicenter retrospective cohort study using data from a French primary care database. Data were collected from January 2012 to January 2022. Glycemic control was estimated by the threshold of glycated hemoglobin and diabetes severity by the number, and the type, of antidiabetic treatments. Infectious risk was evaluated by the mean of antibiotic prescriptions per year.

### Results

Among 59,020 patients, 1959 patients were included in the final analysis. The threshold of glycated hemoglobin was not associated with the mean of antibiotic prescriptions per year (ANOVA p = 0.228). Secondary analyses did not show an association between the number, or the type, of antidiabetic treatments and the mean of antibiotic prescriptions per year (p = 0.53 and p = 0.018, respectively).

No association was observed between glycemic control, diabetes severity and infectious risk in patients with type 2 diabetes. This is the first European study using data from primary care to examine bacterial infectious risk in patients with type 2 diabetes, demonstrating the possibilities offered by the use of databases in primary care research.

### Conclusion

Long-term glycemic control was not associated with bacterial infectious risk in patients with type 2 diabetes.

normandie@univ-rouen.fr, or by contacting the doctor in charge of IT at mikaeldusenne@gmail.com. Other researchers have the option of redoing the analyses, reproducing the studies carried out following a request to the ethical and scientific committee. The application form is available in the rules of procedure of the Ethics and Scientific Committee.

**Funding:** The author(s) received no specific funding for this work.

**Competing interests:** The authors have declared that no competing interests exist

## Introduction

Diabetes affected 537 million people worldwide in 2021 [1] and 5.3% of the French population in 2020 [2]. The management of diabetes is based on the control of glycemia through the implementation of hygienic and dietary rules, oral antidiabetic treatments, insulin treatments and the screening and prevention of microvascular and macrovascular complications [3]. The total amount of reimbursements to people with pharmacologically treated diabetes was estimated at 12.5 billion euros in 2007 in France [4].

In recent meta-analyses [5–8], strict glycemic control by intensification of drug therapies did not lower the risk of macrovascular and microvascular complications with respect to clinically relevant variables: mortality by myocardial infarction, stroke, amputation, renal failure requiring dialysis, blindness, neuropathic pain. The intensification of glucose-lowering therapy is frequently associated with severe adverse events such as hypoglycemia [5,8]. These data give rise to a reflection on a "non-gluco-centric" management of people with type 2 diabetes.

The risk of infection was estimated to be higher in people with type 2 diabetes than in those without diabetes. Notably, with an Odds Ratio (OR) of 1.32 for lower respiratory infections, an OR of 1.24 for urinary tract infections, an OR of 1.33 for bacterial infections of the skin and mucous membranes and an OR of 1.44 for fungal infections of the skin and mucous membranes [9]. This infectious risk is due to immunological changes whose pathophysiological mechanism remains unclear [10]. There is no consensus on the impact of long-term glycemic control on this infectious risk and the results of studies on this subject seem heterogeneous depending on patient inclusion criteria such as hospital or primary care recruitment, duration of diabetes, type of diabetes and definition of glycemic control [11].

The objective of this study was to evaluate whether glycemic control and severity of diabetes were associated with bacterial infectious risk in type 2 diabetes.

## Materials and methods

### Design of the study

We designed a retrospective observational multicenter cohort study using data from a primary care database in Normandy, France.

### Data source

Data were extracted from the electronic medical records of 103,649 patients who had primary care visits with 39 family practitioners in four primary care centers in Normandy. The extracted data concerned primary care visits between January 2012 and January 2022. Data were derived from primary care visits, observations, biometric measurements, history, prescribed treatments, and results of laboratory tests.

### Inclusion of patients

The patients included had to be aged 18 years and older, had type 2 diabetes, and at least three visits in a primary care center. Patients were identified using data from their electronic medical records. We searched International Classification of Primary Care (ICPC) T90 in coded diagnoses or reasons for consultations, labels: "Diabetes II", Type 2 diabetes", "Diabetes mellitus type 2" in history and observations, and Anatomical Therapeutic Chemical (ATC) category A10B which corresponds to antidiabetic treatments in coded prescriptions. Patients who did not have at least three glycated hemoglobin (HbA1c) determinations were excluded from the final analysis.

## Assessment of glycemic control

Glycemic control was estimated by calculating the threshold of HbA1c values of patients with type 2 diabetes. The HbA1c threshold is an average of HbA1c values considering the time between different HbA1c determinations. This method was described by Maple-Brown LJ et al [12] as having a better predictive value than mean HbA1c on the occurrence of microvascular complications. HbA1c thresholds range from < 5.5% to > 9.5%.

## Primary outcome

The primary outcome was an association between the threshold of HbA1c and the mean of antibiotic prescriptions per year. The latter was calculated as the total number of antibiotic prescriptions divided by the number of years of follow-up. Antibiotic prescriptions were identified by searching for the ATC category J01.

## Secondary outcomes

Secondary outcomes were an association between the number, and the type, of antidiabetic treatments and the mean of antibiotic prescriptions per year. We identified all antidiabetic treatments prescribed during the follow-up period, and then the antidiabetic treatments prescribed in the last prescription of each patient using the A10B category of the ATC terminology. These antidiabetic treatments were classified by type of antidiabetic treatment (No antidiabetics, oral antidiabetics, injectable non-insulin antidiabetics, insulin) and by therapeutic class (Biguanides, Sulfonamides, Dipeptidyl Peptidase 4 inhibitors, Glucagon-Like-Peptide-1 receptor agonists, Sodium-Glucose Cotransporter Type 2 inhibitors, Glinides, intestinal-acting antidiabetics, slow-acting insulins, intermediate-acting insulins, rapid-acting insulins and rapid-acting insulin analogues).

Available confounding factors and Hba1c were controlled by a case-control method comparing type 2 diabetic patients who had received at least one antibiotic prescription during follow-up with those who had not.

## Statistical analysis

The number of subjects required was based on the incidence rates of bacterial infections in type 2 diabetic patients with an HbA1c between 5.5 and 6.5% and those with an HbA1c between 8.5 and 9.5% in the Mor et al study (15), for an alpha risk of 5% and a beta risk of 20%, giving a required number of subjects of 276. A first statistical analysis looked for a difference in the mean of antibiotic prescriptions per year by the threshold of HbA1c, and secondly an analysis of the mean of antibiotic prescriptions per year according to the number and type of antidiabetic treatment by ANOVA test and then with a graphical representation by boxplots. The mean of antibiotic prescriptions per year has undergone a logarithmic transformation for reasons of visual clarity of the graphical representation, requiring that the value "zero" be artificially changed to the value 0.0001. Secondly, we performed a multivariate analysis using a negative binomial distribution model, adjusted with HbA1c threshold, age, BMI, history of COPD and asthma, history of cardiac event including stent placement, cancer pathology identified by ICPC codes and text searches in the electronic medical records, number of chronic treatments and number of antidiabetic treatments. The case-control study was carried out using a Student's t test, a Mann Whitney test and a Cohen's d test for quantitative variables, and a chi 2 test for qualitative variables.

### Regulatory

The use of these data was submitted to a scientific and ethical committee: "*Plateforme Régionale d'Information en Medecine Générale*" (PRIMEGE) ethics committee DA-CSE2023-003 and declared to the Commission Nationale Informatique et Liberté (CNIL) as complying with the reference methodology 004 (MR-004) and conforms to the declaration of Helsinki. Patients were informed by posters in the health centers. Data have been fully anonymized, and we had no access to identifying data before, during or after the study. Access to data was requested on 03/14/2023 and was available from the approval of the ethical and scientific committee on 04/14/2023.

## Results

### Description of the population

The database contained the data of 103,649 patients, 84,012 of whom were at least 18 years old at the time of data collection and 72,386 had at least three primary care visits. A total of 59,020 patients met the age and follow-up criteria. We have summarized the process of inclusion and exclusion in a flow chart (Fig 1). The mean year of birth for the overall population was 1973. The male-to-female sex ratio was 0.76 (24,985/32,727). The mean BMI was 27.44 kg/m$^2$.

### Population with type 2 diabetes

Analysis of electronic medical records identified 9247 patients with diabetes: 994 patients by informed diagnosis, 2178 patients by history, 765 patients by reason of visit, 2273 patients by visit data, and 3037 patients by antidiabetic treatments. Combining these methods of detection, 3411 patients (3.29%) were identified as having type 2 diabetes, of whom 1452 patients were excluded because they did not have at least three HbA1c determinations during follow-up. A total of 1959 patients (1.89%) with type 2 diabetes were included in the final analysis. The characteristics of patients with diabetes are shown in (Table 1).

Antidiabetic treatments were prescribed in 90.71% of patients with type 2 diabetes (n = 1777). Non-insulin treatment was prescribed in 65.03% of patients (n = 1274) and insulin treatment in 25.68% (n = 503). The mean threshold of HbA1c of patients with type 2 diabetes was 6.92%. The mean follow-up between the first HbA1c result, and the last primary care visit was 5.32 ± 2.79 years.

### Antidiabetic treatments

Antidiabetic treatments were classified by the number, and the type, of antidiabetic treatments: no antidiabetic treatment, oral antidiabetics, injectable non-insulin antidiabetics, insulin, or a combination. We identified 3750 prescriptions of antidiabetic treatments on the last prescription of patients with type 2 diabetes. The three most prescribed treatments were metformin (37.84%), gliclazide (11.47%) and insulin glargine (10.35%).

### Antibiotic treatments

Antibiotic treatments were prescribed in 1253 (64%) patients (Table 1). We identified 4694 antibiotic prescriptions. Most of them (51.57%) concerned the penicillin class with amoxicillin (31.26%) and amoxicillin/clavulanic acid (15.70%). Macrolides represented 12.46%, streptogramins with pristinamycin 9.16%, quinolones 8.73%, cephalosporins 5.00%, and then phosphonic acids, nitrofurans, sulfonamides associated with diaminopyrimidines and penicillin relatives.

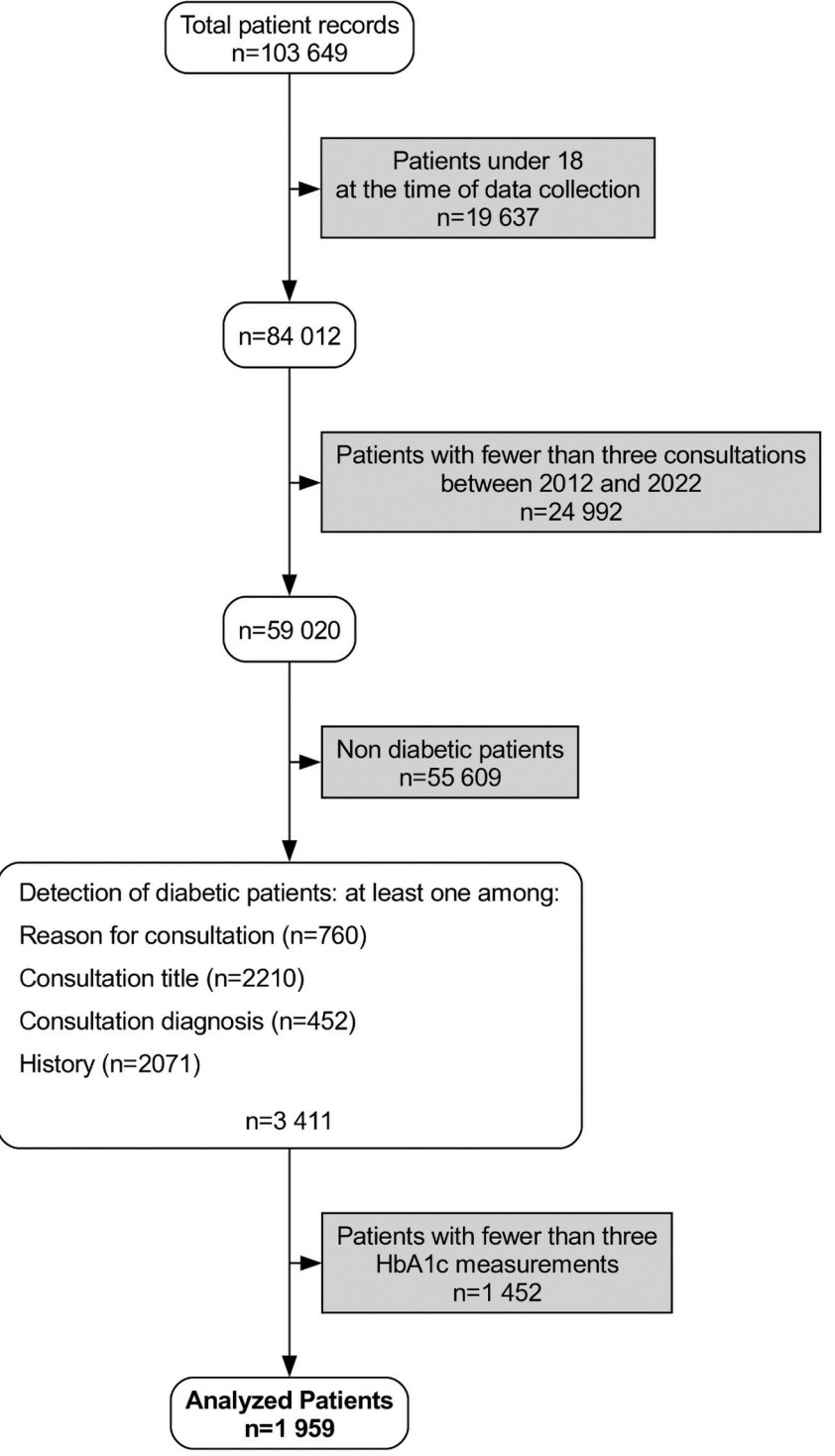

**Fig 1. Flow chart of the inclusion and exclusion process of type 2 diabetic patients.** Hba1c: Glycated hemoglobin.

**Table 1. Characteristics of the overall population and of the population with type 2 diabetes.**

| | Overall population (n = 59 020) | Non diabetic population (n = 57 061) | Population with type 2 diabetes (n = 1959) |
|---|---|---|---|
| Mean year of birth | 1973 ± 20.5 | 1981 ± 22.5 | 1950 ±12.5 |
| Sex ratio (male/female) | 0.76 | 0.75 | 1.24 |
| Mean BMI (kg/m$^2$) | 27.44 | 27.60 | 30.39 |
| Influenza vaccination between 2013 and 2021, n (%) | 843 (1.72%) | 706 (1.24%) | 137 (7.01%) |
| Pneumococcal vaccination between 2019 and 2022, n (%) | 127 (0.26%) | 117 (0.21%) | 10 (0.51%) |
| Covid vaccination between 2020 and 2022, n (%) | 6241 (12.75%) | 5564 (9.75%) | 677 (34.65%) |
| Tobacco use, n (%) | 1959 (4.0%) | 1687 (2.96%) | 272 (13.92%) |
| Antidiabetic treatments No antidiabetic treatment, n (%) | 57 243 (96.99%) | NA | 182 (9.29%) |
| Oral antidiabetics, n (%) | 1041 (1.76%) | NA | 1041 (53.14%) |
| Injectable non-insulin antidiabetics, n (%) | 233 (0.39%) | NA | 233 (11.89%) |
| Insulin, n (%) | 503 (0.85%) | NA | 503 (25.68%) |
| Antibiotic treatments No antibiotic treatment, n (%) | NA | NA | 720 (36.77%) |
| One antibiotic prescription, n (%) | NA | NA | 934 (47.68%) |
| Two antibiotic prescriptions, n (%) | NA | NA | 208 (10.62%) |
| At least three antibiotic prescriptions, n (%) | NA | NA | 111 (5.67%) |

NA: Not Applicable / Not Available, SD: Standard deviation.

## Primary analysis

The primary analysis did not show an association between the threshold of HbA1c and the mean of antibiotic prescriptions per year on ANOVA test (p = 0.228) (Table 2) or between the threshold of HbA1c and the log-transformed mean of antibiotic prescriptions per year on graphical representation (Fig 2).

**Table 2. Antibiotic prescriptions per year according to threshold of HbA1c, and number, and type, of antidiabetic treatments.**

| | | Patients with type 2 diabetes n (%) | Antibiotic prescriptions per year mean ± SD | ANOVA |
|---|---|---|---|---|
| Threshold of HbA1c | • < 5.5% | 27 (1.38%) | 0.88 ± 1.1 | |
| | • 5.5–6.5% | 473 (24.14%) | 0.62 ± 1.14 | |
| | • 6.5–7.5% | 817 (41.70%) | 0.54 ± 1.03 | |
| | • 7.5–8.5% | 422 (21.54%) | 0.50 ± 0.78 | p = 0.228 |
| | • 8.5–9.5% | 158 (8.07%) | 0.57 ± 0.83 | |
| | • > 9.5% | 62 (3.16%) | 0.63 ± 0.83 | |
| Number of antidiabetic treatments | • 0 | 182 (9.29%) | 0.63 ± 1.13 | |
| | • 1 | 897 (45.79%) | 0.53 ± 0.93 | |
| | • 2 | 468 (23.89%) | 0.57 ± 1.11 | p = 0.530 |
| | • > 3 | 412 (21.03%) | 0.59 ± 0.91 | |
| Type of antidiabetic treatments | • No antidiabetic treatment | 182 (9.29%) | 0.63 ± 1.13 | |
| | • Oral antidiabetics | 1041 (53.14%) | 0.49 ± 0.90 | |
| | • Injectable non-insulin antidiabetics | 233 (11.89%) | 0.59 ± 1.04 | p = 0.018 |
| | • Insulin | 503 (25.68%) | 0.65 ± 1.08 | |

SD: Standard deviation.

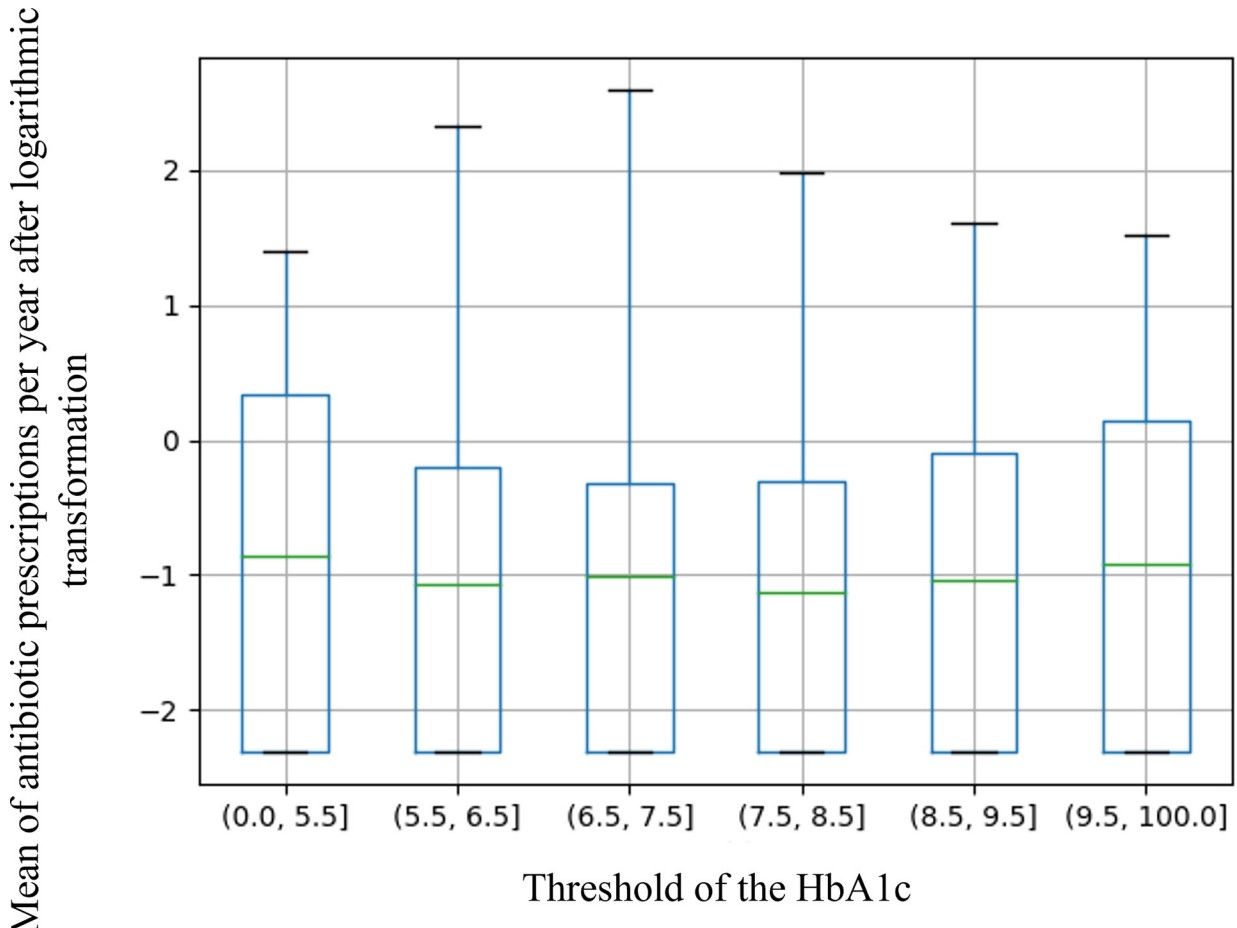

**Fig 2. Boxplot of log-transformed mean of antibiotic prescriptions per year by the categories of threshold glycated hemoglobin.** Glycated hemoglobin (HbA1c) is expressed as a percentage (%).

BMI was not detected in the electronic medical records of 764 patients, who were therefore not included in the multivariate analysis. In multivariate analysis, COPD/asthma was statistically associated with antibiotic prescription with strong coefficients (coefficient 0.60 95% CI [0.383, 0.813] p < 0.001). Number of chronic treatments, and age were statistically associated with antibiotic prescription with a lower coefficient. HbA1c threshold, cardiac history, number of antidiabetics, or malignancy history or BMI were not statistically associated with antibiotic prescriptions. (HbA1c threshold coefficient -0.097 95% CI [-0.207, 0.007] p = 0.070) (Table 3).

## Secondary analysis

Secondary analysis did not show an association between the number, or the type, of antidiabetic treatments and the mean of antibiotic prescriptions per year on ANOVA test (p = 0.53 and p = 0.018, respectively) (Table 2) or boxplot (Fig 3A and 3B, respectively).

The case control showed a significant difference between patients who had at least one antibiotic prescription and those who had not, in terms of sex ratio (1.13 ± 0.13 vs 1.46 ± 0.22 p = 0.007), BMI (31.66 ± 8.65 vs 30.82 ± 7.68, p = 0.048 and Cohen's d-test = 0.12), age, pneumococcal, influenza and Covid 19 vaccination, COPD/Asthma history, malignancy history, number of chronic treatments and number of antidiabetic treatments. There was no

**Table 3. Multivariate analysis of antibiotic prescription per year using the negative binomial distribution model.**

|  | Coefficient | 95% CI | P value |
|---|---|---|---|
| **HbA1c threshold** | -0.092 | [-0.192, 0.007] | p = 0.0688 |
| **Age** | -0.016 | [-0.023, -0.009] | p < 0.001 |
| **BIM** | 0.003 | [-0.013, 0.018] | p = 0.711 |
| **Number of chronic treatments** | 0.077 | [0.058, 0.097] | p < 0.001 |
| **Number of antidiabetic treatments** | 0.003 | [-0.046, 0.053] | p = 0.892 |
| **Cardiovascular history** | 0.203 | [-0.032, 0.439] | p = 0.090 |
| **COPD/Asthma history** | 0.580 | [0.379, 0.781] | p < 0.001 |
| **Malignancy history** | 0.184 | [-0.006, 0.374] | p = 0.057 |

HbA1c: Glycated hemoglobin, BMI: Body Mass Index, COPD: Chronic Obstructive Pulmonary Disease.

significant difference in HbA1c threshold (7.23 ± 1.03 vs 7.17 ± 1.00, p = 0.229), history of tabacco use or cardiovascular history (Table 4).

## Discussion

The main finding of this study was an absence of association between glycated hemoglobin, a surrogate for glycemic control, and the number of antibiotic prescriptions, a surrogate for infectious risk, in patients with type 2 diabetes followed in primary care centers. Multivariate analysis revealed a significant relation between COPD or asthma history and antibiotic prescription. Numerous chronic treatments, which may be an indicator of polymorbidities, and age were also two significant factors, but with a low coefficient, meaning a minor impact on infectious risk. Glycemic control estimated by HbA1c threshold was not associated with a higher number of antibiotic prescriptions during follow-up. These results suggest that the bacterial infectious risk in patients with diabetes is not correlated with chronic glycemic control,

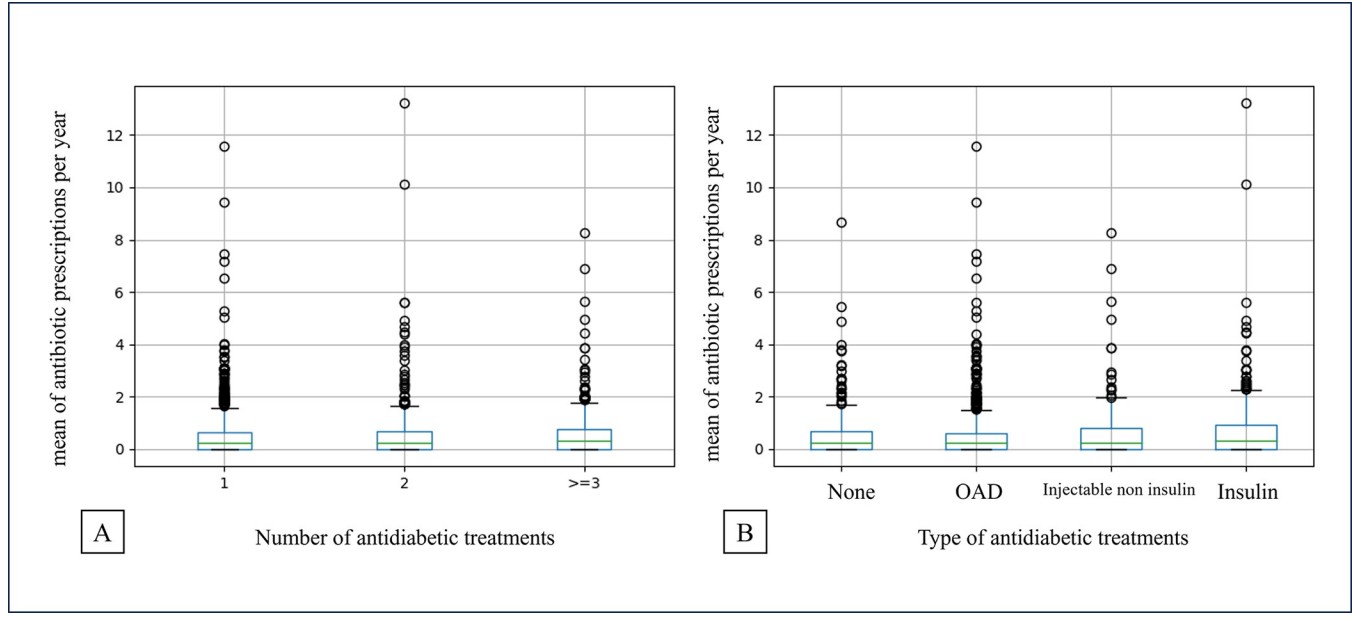

**Fig 3.** (A) Boxplot of the mean of antibiotic prescriptions per year by the number of antidiabetic treatments, (B) Boxplot of the mean of antibiotic prescriptions per year by the type of antidiabetic treatments. OAD: Oral Anti Diabetic treatments.

**Table 4. Cofounding factors and HbA1c according to diabetic patients with at least one antibiotic prescription and diabetic patients with no antibiotic prescription.**

| | Diabetic patients with no prescription (n = 707) | Diabetics patients with at least one prescription (n = 1252) | Statistical analysis |
|---|---|---|---|
| **HbA1c Threshold (%)** | 7.17 ± 1.00 | 7.23 ± 1.03 | p = 0.229[1] |
| **Sex ratio M/F** | 1.46 ± 0.22 | **1.13 ± 0.13** | **p = 0.007[2]** |
| **Age (year)** | 69.11 ± 12.68 | **70.71 ± 12.16** | **p = 0.006[1]** |
| **BMI (kg/m$^2$)** | 30.82 ± 7.68 | **31.66 ± 8.65** | **p = 0.048[1]** |
| **History of tabacco use (%)** | 19.0 ± 2.90 | 17.0 ± 2.10 | p = 0.430[2] |
| **Pneumococcal vaccination (%)** | 3.50 ± 1.40 | **8.80 ± 1.60** | **p < 0.001[2]** |
| **Influenza vaccination (%)** | 14.0 ± 2.60 | **21.0 ± 2.20** | **p < 0.001[2]** |
| **Covid-19 vaccination (%)** | 33.0 ± 3.50 | **37.0 ± 2.70** | **p < 0.001[2]** |
| **COPD/Asthma history (%)** | 3.70 ± 1.40 | **13.0 ± 1.90** | **p < 0.001[2]** |
| **Cardiovascular history (%)** | 7.20 ± 1.90 | 9.10 ± 1.60 | p = 0.148[2] |
| **Malignancy history (%)** | 13.0 ± 2.50 | **21.0 ± 2.30** | **p < 0.001[2]** |
| **Number of antidiabetic treatments** | 2.1 ± 1.61 | **2.58 ± 2.0** | **p < 0.001[1]** |
| **Number of chronic treatments** | 7.27 ± 3.83 | **8.55 ± 4.07** | **p < 0.001[1]** |

HbA1c: Glycated hemoglobin, BMI: Body Mass Index.

[1] Student t test.

[2] Chi 2 test.

however, this risk could be higher in patients with multiple comorbidities, notably pulmonary. Moderate glycemic control with glycated hemoglobin below 9.5% seems to be tolerable in terms of bacterial infectious risk. This risk could be increased during episodes of acute hyperglycemia either by direct glucose toxicity or by its action on the immune cells [13].

Secondary analyses revealed an absence of association between the number, and the type, of antidiabetic treatments, a surrogate for diabetes severity, and the number of antibiotic prescriptions. The case-control study identified confounding factors that could potentially influence infectious risk, namely gender, BMI, COPD/Asthma history, malignancy history, and the number of antidiabetic treatments. Nevertheless, even if there is statistical significance for BMI, Cohen's low d test shows that this factor has a low impact on the risk of infection in our population of type 2 diabetics. Vaccination rates were higher in type 2 diabetic patients with at least one antibiotic prescription. This difference between the two groups is probably not due to the vaccination statue but may be associated with GPs clinical uncertainty, who vaccinate more the patients they consider at risk of bacterial infection. The case control did not reveal any difference in terms of glycemic control, which confirms the results of the primary analysis.

## Comparison with the literature

In an analysis of the UK Clinical Practice Research Datalink (CPRD) database, authors demonstrated a strong association between poor glycemic control and infectious risk, mainly for glycated hemoglobin levels above 11%, with a relative risk of 8.71 for osteoarticular infections, 5.56 for endocarditis, 2.68 for pneumonia, 2.29 for skin infections compared to glycated hemoglobin levels between 6% and 6.5% [14]. This difference in results with our study may be explained by a higher power due to a larger number of patients, especially for extreme values of glycated hemoglobin. Secondly, the authors had access to hospital data to study severe infectious episodes.

A Danish study reported a more moderate association between glycemic control and risk of infection in patients with a mean glycated hemoglobin level of more than 10.5%, with a 1.2-fold increase in the risk of community-acquired infections and a 1.6-fold increase in the risk of hospital-acquired infections. These authors suggested that this risk of infection could be correlated with acute glycemic imbalances, and therefore possibly reversible when glycemic control was restored [15]. These differences in results with our study may be related to several limitations of this study, namely, the lack of identification of patients with diabetes treated by diet alone, and the estimation of glycemic control by a measurement of glycated hemoglobin in the period of acute infection managed in a hospital setting which could overestimate the possible association between chronic glycemic control and infectious risk. Finally, the large number of patients identified with type 2 diabetes increased the power and provided a better estimate of infectious risk in subgroups with extreme values of glycated hemoglobin.

In an analysis of data from the Royal College of General Practitioners Research and Surveillance Center (RCGP RSC), authors did not observe an association between glycemic control and the risk of ocular infections or a difference in infectious events between patients with and without diabetic retinopathy [16]. We report similar results using the same method of HbA1c threshold. The use of this method was of greater interest in this study since it also focused on a microvascular complication, namely diabetic retinopathy. However, this study focused only on ocular infections.

A study on the RCGP RSC database showed a moderately higher risk of infection in patients with type 2 diabetes with moderate glycemic control (HbA1c between 7–8.5%) or poor glycemic control (HbA1c greater than 8.5%) compared to those considered well controlled (HbA1c less than 7%) except for viral and gastrointestinal infections [17]. These results differ from ours because the infectious events were determined by the diagnosis of primary care visits coded in the electronic medical records of patients which allowed them to reach a significant difference for certain infections, in particular for acute bronchitis which does not require antibiotic therapy and for genital and perineal infections represented mainly by genital candidiasis and thus treated by antifungals.

## Strengths of the study

The strengths of this study lie firstly in the analysis of a previously unexplored primary care database, which has enabled the evaluation of a large cohort of patients with type 2 diabetes, representing a real-life research approach. To the best of our knowledge, this is the first European study to examine the global risk of bacterial infection in patients with type 2 diabetes based on data from primary care.

Secondly, this study considers chronic glycemic exposure over several years, estimated by threshold of glycated hemoglobin, and the estimation of glycemic control by categorization into multiple groups. The sample of patients with type 2 diabetes identified in the database can be considered representative.

## Limitations of the study

The main limitation of this study is the lack of power due to the limited number of patients with type 2 diabetes and the number of subgroups of these patients.

Another limitation is the absence in primary care data of information on antibiotic prescriptions in hospitals, where most of severe infectious events are managed [13,14]. Also, the data extraction methods used did not allow us to obtain perfectly precise data. In addition, data were approximate due to their retrospective nature and the fact that they were collected during primary care visits.

Finally, primary care data do not include prescriptions, or information such as history and diagnoses that have not been coded or digitized. In a study in Normandy and Provence Alpes Côte d'Azur French regions, 51.7% of family practitioners declared that they never coded and 19.9% did not identify the treatments prescribed using the software database [18]. Finally, we considered that one antibiotic prescription corresponded to one infectious episode, even though in practice there may be several prescriptions for the same infectious episode. The number of infectious bacterial episodes may therefore have been overestimated, especially as a Dutch study suggested that 25% of infectious episodes led to an appropriate or inappropriate antibiotic prescription, and 7.9 to 9% of antibiotic prescriptions where prescribed additional a first prescription within one infectious disease episode [19]. Furthermore, in an analysis of French antibiotic prescriptions, 50% were unnecessary [20]. Nevertheless, Mor.A et al had demonstrated that type 2 diabetic patients were at greater risk of infectious episodes by observing antibiotic prescriptions in Denmark [21].

## Primary care databases

Beyond the results of this study, this research work highlights the possibilities offered by the analysis of primary care databases. The data collected during primary care visits constitute a vast source of information for clinical or epidemiological research purposes and for improving practices to ensure optimal and increasingly evidence-based care for patients. The use of databases requires continuous improvement of the quality of the data collected during visits and their extraction, by developing automatic coding tools for the medical software of family practitioners [22]. Larger databases such as the RCGP RSC or the CPRD in the United Kingdom demonstrate these research possibilities. For example, the CPRD database contains the data of 16 million patients and has produced 3000 publications since its creation 30 years ago [23], proving its ability to improve the practice and quality of scientific publications in primary care [24].

## Research perspectives

Until now, infectious complications of type 2 diabetes have not been explored either in prospective cohort studies or in randomized clinical trials including patients with type 2 diabetes. Pearson et al. underlined a lack of data in the literature notably on the duration of infection or the severity of diabetes, the impact of the type of diabetes on infectious risk, as well as the antibiotic sensitivities of infectious events [11].

To avoid the biases associated with retrospective analysis, future studies could evaluate the antibiotic consumption of each patient according to their glycemic variations over time. It would be relevant to look for an association between the severity of diabetes and in particular the presence of microvascular and/or macrovascular complications and the risk of infection.

Finally, the pathophysiology of infectious risk in patients with type 2 diabetes remains unclear in the literature. To bridge this gap, clinical and fundamental research is needed to better understand these mechanisms.

## Conclusion

Although we were unable to show an association between long-term glycemic control and bacterial infectious risk, the fact that we used data from a primary care database demonstrates the feasibility of using databases in primary care research.

## Acknowledgments

The authors are grateful to Nikki Sabourin-Gibbs, CHU Rouen, for her help in editing the manuscript.

## Author Contributions

**Conceptualization:** Edouard Lemoine, Matthieu Schuers.

**Data curation:** Mikaël Dusenne.

**Formal analysis:** Mikaël Dusenne.

**Investigation:** Edouard Lemoine.

**Methodology:** Edouard Lemoine.

**Supervision:** Matthieu Schuers.

**Validation:** Mikaël Dusenne, Matthieu Schuers.

**Writing – original draft:** Edouard Lemoine.

**Writing – review & editing:** Edouard Lemoine.

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
