## [Decision Letter · Decision Letter 0]

16 Apr 2024

PONE-D-23-29894Glycemic control and bacterial infectious risk in type 2 diabetes: a retrospective cohort from a primary care database.PLOS ONE

Dear Dr. Lemoine,

Thank you for submitting your manuscript to PLOS ONE. After careful consideration, we feel that it has merit but does not fully meet PLOS ONE’s publication criteria as it currently stands. Therefore, we invite you to submit a revised version of the manuscript that addresses the points raised during the review process.

We look forward to receiving your revised manuscript.

Kind regards,

Peng Gao, Ph.D.

Academic Editor

PLOS ONE

Reviewers' comments:

Reviewer's Responses to Questions

**Comments to the Author**

1. Is the manuscript technically sound, and do the data support the conclusions?

Reviewer #1: Yes

Reviewer #2: No

2. Has the statistical analysis been performed appropriately and rigorously? 

Reviewer #1: Yes

Reviewer #2: No

3. Have the authors made all data underlying the findings in their manuscript fully available?

Reviewer #1: Yes

Reviewer #2: Yes

4. Is the manuscript presented in an intelligible fashion and written in standard English?

Reviewer #1: Yes

Reviewer #2: Yes

5. Review Comments to the Author

Reviewer #1: This study conducted a multicenter retrospective cohort study using data from a French primary care database to evaluate whether glycemic control and severity of diabetes were associated with bacterial infectious risk in type 2 diabetes. Primary analyses showed the threshold of glycated hemoglobin was not associated with the mean of antibiotic prescriptions per year. Secondary analyses did not show an association between the number, or the type, of antidiabetic treatments and the mean of antibiotic prescriptions per year. In conclusion, long-term glycemic control was not associated with bacterial infectious risk in patients with type 2 diabetes.

Question 1：Were sample size calculation carried out? How was it done? This needs to be detailed in the manuscript.

Question 2：There is no consensus on the impact of glycemic control on this infectious risk and the results of studies on this subject seem heterogeneous, lack of information regarding the heterogeneous results.

Question 3：What’s the follow-up criteria of overall population, as “The patients included had to be aged 18 years and older, had type 2 diabetes, and at least three visits in a primary care center” is for population with type 2 diabetes.

Question 4：A flow chart of study identification, inclusion and exclusion criteria will be helpful.

Reviewer #2: This multicenter retrospective cohort study aim to analysis the relationship between glycemic control and diabetes severity and infectious risk. It is the first study to explore the global risk of bacterial infection in T2DM. However, there are several areas that require further clarification.

Major comments:

1. Although the author provided detailed explanations on the inclusion criteria, the exclusion criteria were not clear enough. For example, diabetes patients with other diseases may also increase the risk of infection.

2. The statistical analysis in this article is not very appropriate. In table 1, the overall population should include the population with type2 diabetes, so the description of the antidiabetic treatments is incorrect. It is better to compare the diabetics who had an infection with diabetic case-controls who did not have an infection to ensure consistency between the baselines of the two groups or whether the conclusions are influenced by other factors.

3. Similarly, in Table 2, we still do not know whether the two groups are matched. And whether the mean of antibiotic prescription per year has undergone a logarithmic transformation? Moreover, the “SD value” is greater than the “mean value”, making the data confusing.

4. Some references are in French and English should be used uniformly.

6. PLOS authors have the option to publish the peer review history of their article (what does this mean?). If published, this will include your full peer review and any attached files.

Reviewer #1: No

Reviewer #2: No

---

## [Author Response · Author response to Decision Letter 0]

18 Jun 2024

When submitting your revision, we need you to address these additional requirements. Please ensure that your manuscript meets PLOS ONE's style requirements, including those for file naming. The PLOS ONE style templates can be found at 

The title page has been updated to meet PLOS ONE’s style requirements.

Please provide additional details regarding participant consent. In the ethics statement in the Methods and online submission information, please ensure that you have specified (1) whether consent was informed and (2) what type you obtained (for instance, written or verbal, and if verbal, how it was documented and witnessed). If your study included minors, state whether you obtained consent from parents or guardians. If the need for consent was waived by the ethics committee, please include this information. 

Data were fully anonymized, and we had no access to identifying data before, during and after the study. Under French regulations, retrospective database studies do not require patient consent. They do, however, require information, which has been provided by posters in physicians' offices as recommended by the CNIL Commission nationale de l'informatique et des libertés (French Data Protection Authority). If patients did not wish their data to be used, they could contact the PRIMEGE Normandie data manager to have their data withheld.

We uploaded the Rules of Procedure of the scientific and ethical committee of the PRIMEGE health database. 

We note that you have indicated that there are restrictions to data sharing for this study. PLOS only allows data to be available upon request if there are legal or ethical restrictions on sharing data publicly. For more information on unacceptable data access restrictions, please see http://journals.plos.org/plosone/s/data-availability#loc-unacceptable-data-access-restrictions. 

Access to aggregated raw data is possible through a request to PRIMEGE Normandie's ethical and scientific committee as described in the database's internal regulations, either by contacting the data manager at primege.normandie@univ-rouen.fr, or by contacting the doctor in charge of IT at mikaeldusenne@gmail.com.

Other researchers have the option of redoing the analyses, reproducing the studies carried out following a request to the ethical and scientific committee. The application form is available in the rules of procedure of the Ethics and Scientific Committee.

Reviewer 1 : 

Q1 : Were sample size calculation carried out? How was it done? This needs to be detailed in the manuscript

The number of subjects required was based on the incidence rates of bacterial infections in T2DM patients with an HbA1c of between 5.5 and 6.5% and those with an HbA1c of between 8.5 and 9.5% in the Mor et al study (1), for an alpha risk of 5% and a beta risk of 20%, giving a required number of subjects of 276.

1. Mor A, Dekkers OM, Nielsen JS, Beck-Nielsen H, Sørensen HT, Thomsen RW. Impact of Glycemic Control on Risk of Infections in Patients with Type 2 Diabetes: A Population-Based Cohort Study. American Journal of Epidemiology. 2017;186(2):227 36

Q2 : There is no consensus on the impact of glycemic control on this infectious risk and the results of studies on this subject seem heterogeneous, lack of information regarding the heterogeneous results

Concerning the heterogeneity of results from other studies, results from studies investigating the link between glycemic control and incidence of infection are heterogeneous, with a significant difference depending on whether the study population is from primary care or hospital recruitment, and on the methods or the definition used to determine glycemic control or infectious risk. The study by Pearson et al. (1) identified a need for research into the impact of long-term glycemic control, particularly HbA1c, and infectious susceptibility in diabetic patients. 

We may need to clarify in the sentence you quote "impact of long-term glycemic control" which is justified by the literature review by Pearson et al and detail the criteria by which we judge these studies to be heterogeneous.

1. Pearson-Stuttard J, Blundell S, Harris T, Cook DG, Critchley J. Diabetes and infection: assessing the association with glycaemic control in population-based studies. The Lancet Diabetes & Endocrinology. 1 févr 2016;4(2):148 58

Q3 : What’s the follow-up criteria of overall population, as “The patients included had to be aged 18 years and older, had type 2 diabetes, and at least three visits in a primary care center” is for population with type 2 diabetes.

- The follow-up criteria of the overall population are the same as for type 2 diabetes patients, i.e. be aged 18 years and older and at least 3 consultations at the primary care center.

Q4 : A flow chart of study identification, inclusion and exclusion criteria will be helpful. 

- The flow chart is now done. (Fig 1) 

Reviewer 2 : 

Q1 : Although the author provided detailed explanations on the inclusion criteria, the exclusion criteria were not clear enough. For example, diabetes patients with other diseases may also increase the risk of infection.

You are perfectly right to point out that other chronic diseases are likely to increase the risk of infection (COPD, HIV, heart failure, etc.), as shown by the example of invasive pneumococcal infections (1). We did not adjust for other chronic diseases increasing the risk of infection, and their prevalences were probably significant in the different groups. This is a limitation that we must indeed stipulate in the article and should be taken in consideration in further studies.

The retrospective cohort study design makes it possible to exclude as few patients as possible, with the aim of increasing the statistical power of the study and carrying out a real-life study. 

1. Moe H. Kyaw, Charles E. Rose, Alicia M. Fry, James A. Singleton, Zack Moore, Elizabeth R. Zell, Cynthia G. Whitney, for the Active Bacterial Core Surveillance Program of the Emerging Infections Program Network, The Influence of Chronic Illnesses on the Incidence of Invasive Pneumococcal Disease in Adults, The Journal of Infectious Diseases, Volume 192, Issue 3, 1 August 2005, Pages 377–386, https://doi.org/10.1086/431521

Q2 : The statistical analysis in this article is not very appropriate. In table 1, the overall population should include the population with type2 diabetes, so the description of the antidiabetic treatments is incorrect. It is better to compare the diabetics who had an infection with diabetic case-controls who did not have an infection to ensure consistency between the baselines of the two groups or whether the conclusions are influenced by other factors. 

- For statistical analysis, the ANOVA test can be used to determine whether at least one of the groups has a statistically different result from the others, which is not the case in our study. Statistical analysis of multiple groups is made difficult here by the heterogeneity of the size of the different groups.

- Concerning table 1, We have modified the table by adding a 3rd column including non-diabetic patients for greater clarity and corrected the Overall Population column. However, we did not search for anti-diabetic treatments in non-diabetic patients, so the proportions of the number of treatments are probably wrong in this population. Perhaps we should therefore leave the term NA for these results.

- On your recommendation, we carried out a case-control study of diabetic patients who had received at least one antibiotic prescription, compared with those who had not. The results enabled us to identify certain potentially confounding factors, such as gender and BMI, although further statistical analysis showed their impact to be low. 

In addition, glycated hemoglobin was not different between the two groups, supporting the main findings of the study. We add a table with results. (Table 3) 

Q3 : Similarly, in Table 2, we still do not know whether the two groups are matched. And whether the mean of antibiotic prescription per year has undergone a logarithmic transformation? Moreover, the “SD value” is greater than the “mean value”, making the data confusing. 

- The population studied is composed solely of identified type 2 diabetics, who have been separated and analyzed into several groups according to their glycemic control, the number of antidiabetic treatments and the type of antidiabetic treatment, without any matching between the different groups.

- Concerning the logarithmic transformation, it only concerns the graphical representation in boxplot for reasons of visual clarity.

- Regarding standard deviations that are higher than the mean, this can be explained by the fact that most values are quite low with a long tail of high values (skew). This can be seen with values that can only be positive, with a concentration towards zero, and whose distribution follows a multiplicative mode. 

Q4 : Some references are in French and English should be used uniformly.

References have been translated.

Thank you again for your interest in our study and for helping to improve it. You have our gratitude.

If you have any further questions or comments, please do not hesitate to contact us.

Dr Lemoine Edouard

---

## [Decision Letter · Decision Letter 1]

15 Jul 2024

PONE-D-23-29894R1Glycemic control and bacterial infectious risk in type 2 diabetes: a retrospective cohort from a primary care database.PLOS ONE

Dear Dr. Lemoine,

Thank you for submitting your manuscript to PLOS ONE. After careful consideration, we feel that it has merit but does not fully meet PLOS ONE’s publication criteria as it currently stands. Therefore, we invite you to submit a revised version of the manuscript that addresses the points raised during the review process.

We look forward to receiving your revised manuscript.

Kind regards,

Peng Gao, Ph.D.

Academic Editor

PLOS ONE

Reviewers' comments:

Reviewer's Responses to Questions

**Comments to the Author**

1. If the authors have adequately addressed your comments raised in a previous round of review and you feel that this manuscript is now acceptable for publication, you may indicate that here to bypass the “Comments to the Author” section, enter your conflict of interest statement in the “Confidential to Editor” section, and submit your "Accept" recommendation.

Reviewer #2: All comments have been addressed

Reviewer #3: (No Response)

2. Is the manuscript technically sound, and do the data support the conclusions?

Reviewer #2: Partly

Reviewer #3: No

3. Has the statistical analysis been performed appropriately and rigorously? 

Reviewer #2: Yes

Reviewer #3: No

4. Have the authors made all data underlying the findings in their manuscript fully available?

Reviewer #2: Yes

Reviewer #3: Yes

5. Is the manuscript presented in an intelligible fashion and written in standard English?

Reviewer #2: Yes

Reviewer #3: Yes

6. Review Comments to the Author

Reviewer #2: 1.Please check the spelling of HbAlc in Table 3.

2.Please make sure that the ways of the representation of continuous variables （mean（SD）or mean ± SD） are consistent in Tables 1, 2, and 3.

Reviewer #3: Despite revising most of the review comments, the authors did not address two key issues. Firstly, the potential impact of other major diseases was not ruled out, especially considering the possibility that widespread neocoronary epidemics could elevate the risk of bacterial infections. Secondly, there was a lack of baseline comparison between diabetic patients without infections and those with infections. Furthermore, age and duration of diabetes are significant factors that may affect bacterial infection, yet no corrective analysis was conducted to account for these variables.

7. PLOS authors have the option to publish the peer review history of their article (what does this mean?). If published, this will include your full peer review and any attached files.

Reviewer #2: No

Reviewer #3: No

---

## [Author Response · Author response to Decision Letter 1]

3 Oct 2024

Dear reviewers, 

Thank you again for your interest in our article and for your comments and suggestions, which have greatly improved its quality. 

Reviewer 1:

1 Please check the spelling of HbAlc in Table 3.

We have harmonized all HbA1c abbreviations.

2.Please make sure that the ways of the representation of continuous variables （mean（SD）or mean ± SD） are consistent in Tables 1, 2, and 3.

All standard deviations have been harmonized in the tables.

Reviewer 2: 

Despite revising most of the review comments, the authors did not address two key issues. Firstly, the potential impact of other major diseases was not ruled out, especially considering the possibility that widespread neocoronary epidemics could elevate the risk of bacterial infections. Secondly, there was a lack of baseline comparison between diabetic patients without infections and those with infections. Furthermore, age and duration of diabetes are significant factors that may affect bacterial infection, yet no corrective analysis was conducted to account for these variables.

To improve identification of confounding factors, we supplemented the case-control section with other variables: Vaccination rate, smoking status, number of chronic treatments on the usual prescription, number of antidiabetic treatments, history of malignancy, COPD/Asthma or Cardiovascular event. The results have been presented in table 4.

We also performed a multivariate analysis using a negative binomial model, taking into consideration BMI, age, smoking status, HbA1c threshold, history of asthma and COPD, cardiovascular event and history of malignancy. This required us to design new pathology detection algorithms in the database.

This analysis enabled us to confirm our results and to identify confounding factors and the strength of their impact on the number of antibiotic prescriptions. The results are presented in Table 3. 

Concerning the duration of diabetes, this is an important factor which is difficult to estimate in our database, as it is poorly structured. Nevertheless, this factor could be studied in the future, once the database has been linked with health insurance data.

Thank you for your comments, we hope we've answered them well.

Dr Lemoine Edouard, MD

---

## [Decision Letter · Decision Letter 2]

8 Nov 2024

Glycemic control and bacterial infectious risk in type 2 diabetes: a retrospective cohort from a primary care database.

PONE-D-23-29894R2

Dear Dr. Lemoine,

We’re pleased to inform you that your manuscript has been judged scientifically suitable for publication and will be formally accepted for publication once it meets all outstanding technical requirements.

Kind regards,

Peng Gao, Ph.D.

Academic Editor

PLOS ONE

Additional Editor Comments (optional):

Reviewers' comments:

Reviewer's Responses to Questions

**Comments to the Author**

1. If the authors have adequately addressed your comments raised in a previous round of review and you feel that this manuscript is now acceptable for publication, you may indicate that here to bypass the “Comments to the Author” section, enter your conflict of interest statement in the “Confidential to Editor” section, and submit your "Accept" recommendation.

Reviewer #2: All comments have been addressed

Reviewer #3: All comments have been addressed

2. Is the manuscript technically sound, and do the data support the conclusions?

Reviewer #2: Yes

Reviewer #3: Yes

3. Has the statistical analysis been performed appropriately and rigorously? 

Reviewer #2: Yes

Reviewer #3: Yes

4. Have the authors made all data underlying the findings in their manuscript fully available?

Reviewer #2: Yes

Reviewer #3: Yes

5. Is the manuscript presented in an intelligible fashion and written in standard English?

Reviewer #2: Yes

Reviewer #3: Yes

6. Review Comments to the Author

Reviewer #2: The authors have adequately addressed the comments raised in a previous round of review and I feel that this manuscript is now acceptable for publication.

Reviewer #3: (No Response)

7. PLOS authors have the option to publish the peer review history of their article (what does this mean?). If published, this will include your full peer review and any attached files.

Reviewer #2: No

Reviewer #3: No

---

## [Editor Report · Acceptance letter]

15 Nov 2024

PONE-D-23-29894R2 

PLOS ONE

Dear Dr. Lemoine, 

I'm pleased to inform you that your manuscript has been deemed suitable for publication in PLOS ONE. Congratulations! Your manuscript is now being handed over to our production team.

Kind regards, 

on behalf of

Professor Peng Gao 

Academic Editor

PLOS ONE